# Improving outcomes for primary school children at risk of cerebral visual impairment (the CVI project): protocol of a feasibility study for a cluster-randomised controlled trial and health economic evaluation

Cathy Williams ![ORCID], Anna Pease ![ORCID], Trudy Goodenough, Katie Breheny, Daisy Gaunt, Parisa Sinai, Rose Watanabe

► http://dx.doi.org/10.1136/bmjopen-2020-044856

Bristol Medical School, University of Bristol, Bristol, UK

**Correspondence to**
Dr Cathy Williams;
cathy.williams@bristol.ac.uk

## ABSTRACT

**Introduction** Cerebral visual impairment (CVI) refers to a spectrum of brain-related vision problems. CVI is associated with poor educational and mental health outcomes. An intervention has been developed to help children with CVI, involving input from clinicians, teachers and parents. The effectiveness of this intervention needs to be evaluated. This study aims to guide any refinements to the intervention or the design of a future cluster-randomised trial that may be needed.

**Methods and analysis** This study will include all methods anticipated for a future cluster-randomised controlled trial. Eight primary schools will be recruited and randomised to receive the intervention or carry on with usual practice. The intervention will comprise an information pack for schools and access to a local paediatric ophthalmology clinic (who are prepared to assess them for CVI), for up to 5% of participating children. Outcome assessments will be carried out at baseline (before randomisation) and after 4–5 months of intervention period. Assessments will include children's self-reported quality of life, their learning ability and behaviour as reported by teachers, and family functioning reported by parents. Cost data will include service use, family expenditure on additional support (eg, private appointments and administration) and school spending and resource used in helping children with special educational needs or disability. A process evaluation (PE) will collect additional data relating to the implementation of the intervention and the trial processes, in the school and clinic settings. The protocol for the PE will be reported separately.

**Ethics and dissemination** Ethical permission was obtained from the University of Bristol Faculty of Health Sciences Ethical Committee. The results will inform the design of a future trial to assess the effectiveness and cost-effectiveness of the intervention and will be shared with participants, CVI-support groups and peer-viewed journals.

**Trial registration number** ISRCTN13762177; Pre-results.

<div style="border:1px solid #000">

### Strengths and limitations of this study

► All elements/methods proposed for the future cluster-randomised controlled trial will be included, which will allow refinements to be made if needed.

► A detailed, mixed methods process evaluation is embedded.

► Children with cerebral visual impairment are not individually identified and the primary analysis will be instead a group presumed to be enriched with affected children, which will limit the statistical power.

► The duration of the study is limited to two terms so we cannot estimate long-term changes in implementation or effect.

► There is no provision to interview general practitioners to understand their views on the suggested referral pathway.

</div>

## INTRODUCTION

Damage or poor function in the brain areas related to vision leads to brain-related visual impairment, also known as cerebral visual impairment or impairments (CVI).[1] There is not yet consensus on the specific type or severity of vision problems that should warrant a diagnosis of CVI, but there is general acceptance that brain-related vision disorders are important and a major cause of registrable blindness in the UK and other developed countries.[2] A recent review suggested CVI was a verifiable visual impairment not attributable to ocular or optic nerve problems.[3] Children with CVI may not be identified if they have good visual acuity;[4] hence are not detected in visual screening programmes recommended for reception year (age 4–5 years) children.[5] They are therefore at risk of poor outcomes because of lack of support.

Simple strategies can be helpful for children with CVI, including decluttering presentations, using a window to isolate part of the page, or using plain backgrounds.[6] Decluttering can also increase engagement with lessons for children without known problems.[7] However, there are no recommendations for teachers regarding CVI or vision problems in general and if these have been diagnosed, the child is usually referred to a specialist teacher (a qualified teacher for the visually impaired (QTVI)).

In a recent study by our group, at least 3.4% of all the children in primary schools we surveyed had CVI-related vision problems.[8] We found that none of the potential indicators of higher risk for CVI-related vision problems (red flags) we examined were very accurate: having extra educational help was 79% sensitive but only 49% specific, while by contrast a vision-specific questionnaire was 91% specific but had only 13% sensitivity.

### Development of the complex intervention

We have developed a multilevel intervention that aims to improve outcomes for children with CVI. The intervention involves sharing information about the condition with the teachers, empowering them to try relevant strategies for children who are struggling and improving links between schools and the paediatric ophthalmology clinic.

Our previous study[8] has suggested that on average at least one child per class has CVI-related vision problems and that the questionnaires we used do not accurately identify affected children in a community setting. The intervention therefore emphasises to teachers that universal approaches (eg, decluttering classrooms) will be helpful, as well as additional targeted approaches for children who are struggling. In this way, all children are exposed to potentially helpful 'vision-friendly' surroundings and there is less risk of a child getting no support because they are not yet diagnosed. A tiered approach using universal as well as targeted support has been recommended for other conditions affecting children, including anxiety and mental health problems.[9 10]

Families have told us that referrals to the eye clinic and treatment there can be difficult if there is little understanding of CVI. Another aspect of the intervention was therefore to provide an enhanced interface between the school and the eye clinic, by ensuring that the clinic was equipped and willing to accept referrals of children who may have CVI.

The school intervention pack materials were complied with advice from our professional advisory group (see below) and our local authority QTVIs and comprise the following:
1. A short (30 min) presentation with audio commentary by the study principal investigator (CW) giving an overview of CVI and how to help affected children and of the other materials provided, plus a transcript of the presentation.
2. Links to existing online resources about CVI.
3. A 'Toolkit'—a word document detailing simple, inexpensive strategies (a) for universal use with whole class

or school and (b) targeted measures to help individual children who are struggling with learning. These were reviewed and amended by our local QTVIs.
4. Some teaching resources (stories and suggested activities for children to do), matched to the curriculum, for teachers to use to help explain topics related to vision and light.
5. Guidance for teachers as to which children may have CVI and which may benefit from a referral to the eye clinic including suggested referral criteria.
6. Letters for teachers to give to parents to show their general practitioners (GPs), requesting referral to the local paediatric ophthalmology clinic because vision problems are suspected. At the request of the University Faculty Ethical Committee, the number of study referrals was capped at 5% of participating children (to avoid overburdening the GPs or the eye clinic) although GPs can make whatever referrals they want outside of the study, in the usual way.

The eye clinic intervention materials comprise the following:
1. A validated questionnaire to score risk for CVI in at-risk children.[11]
2. A protocol for vision tests to identify CVI-related vision problems, as used in our earlier study.[8]
3. Sample advice sheets on how to help children with CVI, adapted from existing websites and resources[12 13] augmented with input from our local QTVIs.
4. A link to a secure Research Electronic Data Capture (REDCap, v9) database to enter the results of the vision tests and print out a summary report to give the families, schools and for the clinic notes.

As part of the process described by the Medical Research Council for the development of complex interventions,[14] we need to evaluate this intervention and see if in real life, the intervention might need changing or adapting.

When and if we have evidence that the intervention is helpful, we will make the study materials including the advice sheets, available on request on our website (www.thecviproject.co.uk).

### How the intervention might work?

There are several pathways by which the intervention might potentially improve outcomes for children with CVI, for example, teachers adapting their teaching methods, or accurately identifying and referring affected children to the eye clinic. We have conceptualised these in a logic model and designed a process evaluation (PE) to collect data on what changes in school processes or attitudes if any, actually occur. We report the protocol for this separately (in press BMJ Open).

### Aim and objectives of this feasibility cluster-randomised controlled trial (cRCT)

The aim of this study is to assess the feasibility and acceptability of the new intervention and of the proposed trial methods, to inform the development of a future fully powered cluster-randomised controlled trial.

**Table 1** Assessment tools to be used

| Respondent | Concept being investigated | Tool | Format |
|---|---|---|---|
| Child | Quality of life | PedsQL Generic Core Scales: Child report: 8–12 years or child report: 5–7 years (supported by adult report) | Paper, in the classroom and supported by school staff Which form the child completes is at discretion of their teacher |
| Teacher | Child learning ability | PedsQL cognitive scales | Online REDCap form— one per child |
| | Child behaviour at school | SDQ with impact supplement | Online REDCap form—one per child |
| | Child behaviour that may indicate CVIs | 5-item CVI questionnaire | Online REDCap form—one per child |
| | Teacher perception of their own teaching | Self-efficacy | Online REDCap form—one per teacher |
| Parents | Parental quality of life related to family | PedsQL family impact module | Paper Parents fill in one per child in the study |
| | Generic and child-related costs and resource input | Family expenses questionnaire (designed for the study) | Paper Parents fill in one per child in the study |
| | Child behaviour that may indicate CVIs | 5-item CVI questionnaire | Paper Parents fill in one per child in the study |

CVI, cerebral visual impairment; PedsQL, Pediatric Quality of Life Inventory TM; REDCap, Research Electronic Data Capture; SDQ, Strengths and Difficulties Questionnaire.

Our specific objectives are to collect data on the following:

Intervention-related outcomes as follows:

1. How the intervention is implemented in practice?—a detailed PE has been designed to do this as reported in a separate protocol paper (ref).
2. Any changes in the outcome data (see table 1) that may indicate an effect of the intervention, to help us calculate the sample size needed for the future trial.
3. Attitudes of teachers and parents to the intervention and whether it is beneficial.

Study design-related outcomes as follows:

We will document the following:

1. The response rate and yield from recruitment flyers, letters and emails sent, any reasons given for non-participation and retention of schools once recruited (once they have signed the study paperwork—see below).
2. The proportion and demographic predictors of parental non-agreement to data sharing in schools that have been recruited.
3. The completeness of data from (a) child-report, (b) teacher-report and (c) parent questionnaires.
4. The number of children suggested by the schools for a referral to the eye clinic, the proportion that attend the eye clinic and of those, the proportions who do and do not have any CVI-related vision problems.

## METHODS AND ANALYSIS
### Design
The design of this feasibility cRCT is shown in the flowchart in figure 1. The processes for children referred to the eye clinic as part of the intervention are shown in figure 2.

### Setting
The study will involve primary schools in three areas of England, UK: Gloucestershire, Somerset and Southampton. These areas were chosen because the local paediatric ophthalmologists have agreed to participate in the study. The study will be led by the study team in Bristol, UK.

### Inclusion criteria
All mainstream state-funded primary schools including academies, local authority-maintained schools and free schools, within 1 hour of travel time to the local ophthalmology unit, in the three study areas will be eligible for inclusion.

### Exclusion criteria
Schools who share a special educational needs and disability coordinator (SENCo or SENDCo), where contamination between intervention to control schools might occur, unless they are part of a linked group of schools who all participate in the trial (and will be in the same arm).

### Enrolment of schools
All schools in the study areas that meet the inclusion criteria will be sent an invitation letter and copies of the school and parent/carer information sheets. Invitation letters and information sheets will be sent out by post and email to heads, SENCos and heads of governors. The letters and information sheets will explain the study and that schools will be reimbursed for the teachers' time completing outcome assessments.

### Legal basis for conducting the study
The legal basis for the study will be the University of Bristol undertaking a public task, as described in articles

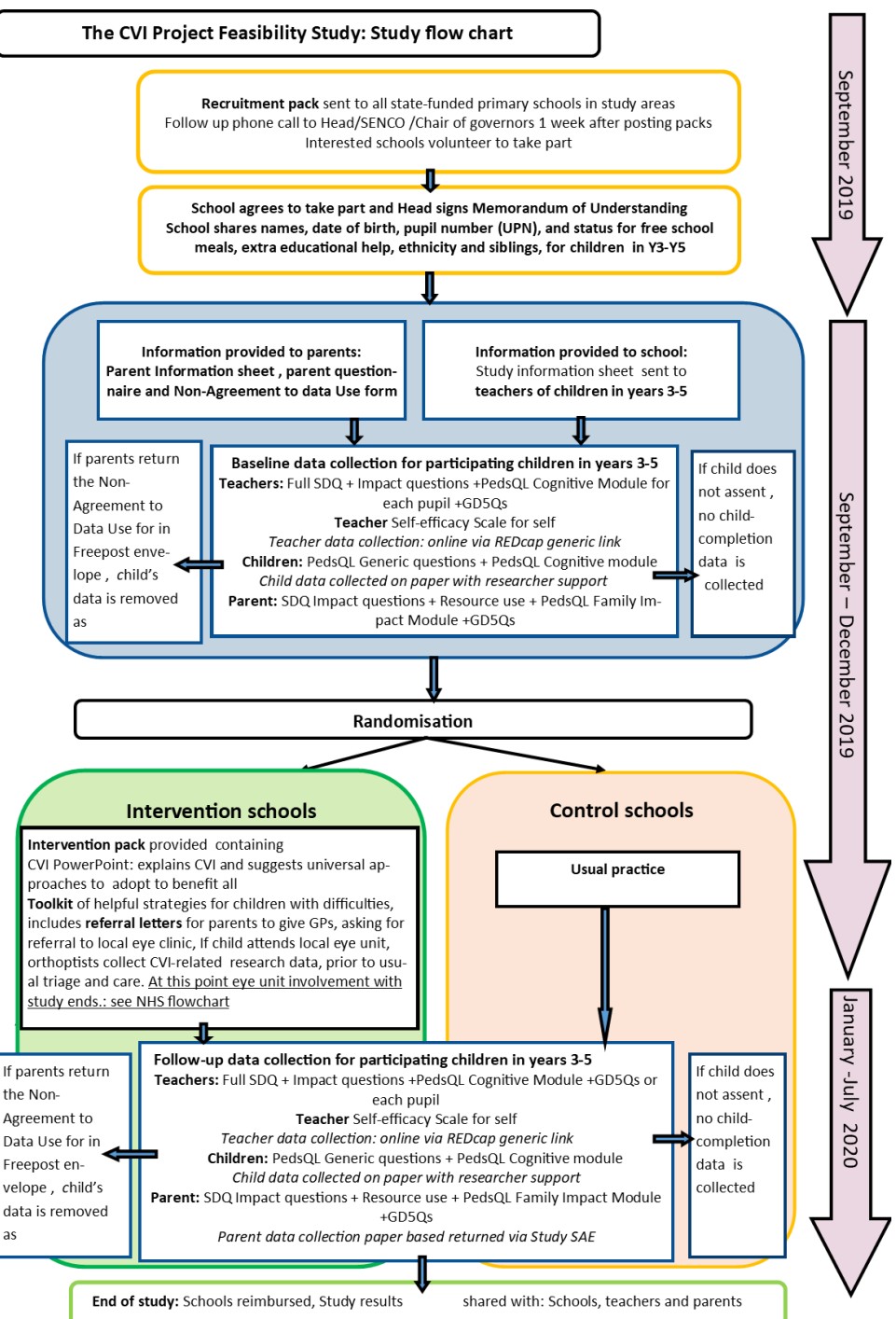

**Figure 1** The CVI project feasibility trial flowchart. CVI, cerebral visual impairment; GP, general practitioner; REDCap, Research Electronic Data Capture; SDQ, Strengths and Difficulties Questionnaire; SENCO, special educational needs and disability coordinator.

6.1e and 9.2j of the European Union General Data Protection Regulation (EU-GDPR). It will not be on the basis of parental consent.

The head teacher will agree to participate on behalf of the children and staff in their school and will sign a Memorandum of Understanding to this effect with clear descriptions of the expectations and obligations of the study team and the participating school. This will be non-binding and is written to foster mutual understanding of

what is involved—there will be no steps taken if schools are unwilling or unable to carry out all their activities. A link to the full privacy notice online will be provided to all parents and school staff and a summary of the privacy notice is included in the parent/carer information sheet (PIS) and the school information sheet.

Any parent in a participating school will be able to withdraw their child's data from the study. This can be achieved by contacting the study team using details

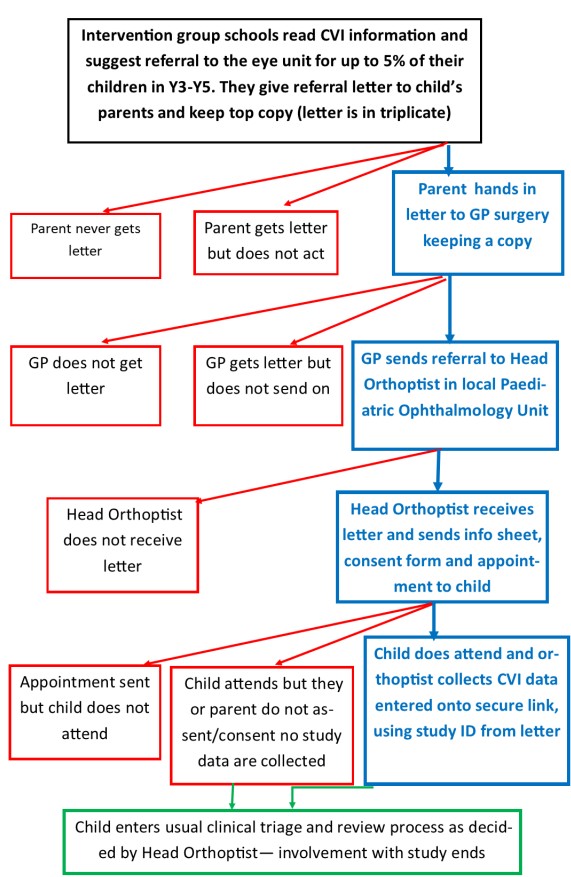

**Figure 2** The process of referral to the eye clinic for intervention schools. CVI, cerebral visual impairment; GP, general practitioner, ID, identification.

included in the PIS, or by completing a non-agreement to data use form that will all be sent with a PIS, plus a freepost envelope. They will be able to do this at any time up to the end of the study. In addition, if a child does not assent to filling in the child-report questionnaires, they do not have to. They will still be included in the study unless their parent has withdrawn them.

### Study set-up and baseline assessments

An excel sheet will be completed by the school with name, gender, date of birth, school pupil number, status regarding free school meals, siblings at the school and the level of and reason for extra educational help, for all children in Y3–Y5. Unique study identification (ID) numbers for each child will be generated. The study team will then provide a PIS and a questionnaire for each family, plus self-completion forms for the children, all with individual study ID numbers.

Outcome assessments were selected from the literature suggesting that unsupported CVI can impair children's learning[15 16] and from a core outcome set study identifying outcomes relating to CVI in children that were rated as important by families and professionals (paper in preparation). The assessment tools to be used

are summarised in table 1. The children will be asked to complete the child-report Pediatric Quality of Life Inventory TM(PedsQL) quality of life (QoL) scales.[17] The teachers will be asked to complete the PedsQL cognitive scales which estimate the children's ability to learn[18] and Goodman's Strengths and Difficulties questionnaire.[19] The parents will be asked to complete the PedsQL Family Impact Module[20] and a resource use questionnaire developed for the study (see below and a copy is given as online supplemental material). Both teachers and parents will also be asked to complete a short vision-specific questionnaire which elicits behaviours suggestive of CVI.[21] Finally, the teachers will be asked to complete a questionnaire about their perception of their own teaching skills and abilities.

The study team will send or deliver to schools all the materials they need (school information sheet and PIS, child questionnaires and parent questionnaires). The parent and child questionnaires will all have a front sheet giving the child's name, class and study ID number, while the actual questionnaires will have only the study ID. On completion, the front sheets will be removed and the data returned to the study team with study ID only, by freepost envelope for the parent questionnaires and child-report forms will be collected from the school by a member of the study team.

Teachers will get a link to an online questionnaire generated by a REDCap database and will complete one for each child in their class, plus answer the self-efficacy questionnaire relating to their own teaching practices. A schedule showing when the assessments will be carried out is shown in online supplemental material.

### Randomisation

After baseline data are collected, a statistician outside the study will randomise the participating schools, with two stratification variables (a) ≤15% children having SEND versus >15% children having SEND and (b) three or more classes per year versus two or fewer classes per year.

The schools will be informed of their allocation by email and telephone call, all on the same day.

### Postrandomisation
#### Study schools

Schools allocated to the intervention group will be sent the CVI intervention pack (by post and by email) and the referral letters to use if they want to suggest a child is seen in the eye clinic. They will be asked to study these as early as possible; however, the actual timing and use of the packs will be examined in detail in the PE.

#### Control schools

Schools allocated to the control group will be asked to carry on as normal.

### Duration of study period

For pragmatic reasons (limited availability of time and funds), the intervention period (the length of time between giving the schools the intervention pack and

collecting the outcome data) will be of two terms; whereas, in the full trial it is expected that the intervention period would be a full year.

### Health economic evaluation

It is anticipated that the future trial will involve a cost-utility analysis, from a public sector perspective that includes family, education and health service costs. We will collect data from parents on their children's use of NHS primary and secondary care resource use and any additional costs incurred to support their child's health and educational needs. This will include time spent pursuing specialist appointments and other administrative tasks as well as costs related to their child's vision. A copy of this questionnaire is included as a online supplemental file.

We will collect data from schools about the resource they allocate to supporting children with SEND, in terms of numbers and hours of learning support assistants, costs of any special equipment and requests for external specialists for example, speech and language therapists. We will collect data from the eye clinics about numbers of study children who attend after referral by the teachers, the staff involved, the average time for appointments and the cost of equipment needed. The costs of delivering the intervention by schools will be explored in the process evaluation.

### Blinding and contamination

It will not be possible to blind school staff as to which arm of the study they are in, or the fieldwork staff, however analysis of the quantitative data will be carried out by a separate statistician who is blind to group allocation. The primary outcome for the definitive trial is planned to be the child's self-reported QoL which should be subject to less allocation bias than the teacher-report would be. The feasibility of collecting the child-report data is therefore an important outcome from this study. The intervention group schools will be asked not to share the intervention materials; however, it is recognised that contamination between arms may occur and the PE will investigate this. If there is sharing of information by schools or by parents, we will note this and decide what changes should be made to the parent information leaflets provided in the future trial.

### Sample size

As this is a feasibility cRCT, formal power calculations are not appropriate. Based on our previous study,[8] we aim to recruit adequate primary schools so that 1300 children are involved. Assuming data returns of 95% and a prevalence of 3.5% of CVI-related vision problems, this would give approximately 22 children with CVI-related vision problems each arm, most of whom are expected to be in the group of children having extra educational help. Depending on class sizes and structure, we estimate eight schools will be recruited and randomised.

### Analysis

Analyses will be mainly descriptive and will include 95% CIs wherever appropriate. For between arm comparisons, an intention-to-treat approach will be used.

### Recruitment

We will report the numbers of flyers and invitation letters sent by post and email, and the yield from these to see which produces the highest number of schools agreeing to take part. We will report the numbers and proportions of children whose parents elect not to share their data. We will compare these results with those published by other trials involving primary schools and consider whether any changes in approach would be needed for a full trial.

### Outcome assessment data

We will report the numbers and proportions of participating children for whom we receive child-report, teacher-report and parent-report outcome data at baseline and at follow-up. We will report the proportion of missing data within returned questionnaires.

We will examine and describe all the questionnaire data from children, teachers and parents. Although this is a feasibility study rather than the definitive trial, it will be useful to see the direction and magnitude of any change in the QoL or learning ability scores during the study. We do this for all children and for the subgroup having extra educational help, who we expect will have a higher prevalence of CVI-related vision problems based on our prevalence study in which 42% of children having extra help had one or more CVI-related vision problems.[8] Therefore, we expect this group to be on average more sensitive to the effects of any vision-enhancing interventions they receive. As we cannot be sure who has CVI or CVI-related vision problems without examining all the participants, which would be unfeasible in real life, we will consider the group having extra educational help as a sample that is enriched with children who have CVI. We will also look specifically at the outcome data for any intervention group children referred to the eye clinic and diagnosed with CVI. We will estimate the number of children referred who do not have any CVI-related vision problems, as an indicator of how accurate the teacher referrals are.

We will use these data to guide the plans for the design and analysis of the future trial and estimate the numbers that will be needed for the trial to have appropriate statistical power.

### Health economic evaluation

The health economic analysis will also be predominantly descriptive. Item-level missing data and the return of adult and child questionnaires will be explored. As the intervention is predominantly delivered outside of the NHS, data on individual level resource use in this study is mostly reliant on parent report. The data quality will be explored to inform the refinement of questionnaires and utility of pursuing linked data collection in a future trial.

Completion and return of school and NHS eye clinic data will also be examined.

There is extensive literature[22] mapping the PedsQL to the CHU9D, so we will estimate quality-adjusted life years associated with any change in QoL. The possibility of alternative frameworks including broader outcomes, such as cost-consequence analyses, will also be explored.

## Governance

The study will be managed day-to-day by the study team. Expert advice will be available from the study advisory groups. A Trial Steering Committee (TSC), chaired by an independent expert academic, will oversee the conduct of the study. The TSC will discuss the study results and findings with the study team and will advise on the final decision of whether to apply for funding for a full trial, with modifications to the intervention or study methods if indicated.

## Patient and public involvement

Our parents advisory group (comprising parents of children with CVI) have advised on the design and materials for the study at several timepoints. At the outset they agreed that informing teachers about CVI and how to help affected children was an important intervention. They recommended including outcomes related to child learning ability as well as well-being. They recommended that the control group of schools be offered the intervention materials at the end of the study and that the schools would need reimbursing for the teachers' time spent filling in questionnaires.

At subsequent meetings they revised the wording of the PIS. They commented on the assessment tools (table 1) and recommended that parents might find some questions off-putting so we agreed that using just the Strengths and Difficulties Questionnaire (SDQ) impact module was best for parents. Parents also suggested the term 'vision -friendly' for measures like decluttering. They suggested emphasising to teachers the low cost and ease of the simple measures suggested and that they would be useful as evidence for Ofsted (the national schools' regulatory body), of interventions the teachers were using to support learners. They chose the family impact module above another questionnaire as a tool for capturing the family perspective.

We also have a professionals advisory group (a QTVI, a community paediatrician, an educational psychologist, a head teacher, a SENCo, an orthoptist and a public health consultant). This group have advised regularly on the study materials and design, specifically on the approach to schools, the scheduling of recruitment, the wording of the intervention pack materials and the referral process for children to get to the eye clinic.

## ETHICS AND DISSEMINATION

Ethical approval for the protocol of this study was obtained from the University of Bristol Faculty of Health

Sciences Ethics Committee (FREC ref: 89144) in August 2019. The committee requested changes to the protocol when first submitted: giving parents more detail about CVI in the PIS and capping the number of referrals to the eye clinic to prevent the GPs or eye clinic being overburdened. Once we had made these changes, a favourable opinion to the study was granted.

The study is registered on the International Standard Randomised Controlled Trial Number registry. Approval from the Health Research Authority (HRA-Ref 19/HRA/6124) has been obtained.

Results from this study will be submitted to peer-reviewed journals and presented at scientific conferences. Summaries will be made available to participating schools, our advisory groups and family support group websites and social media platforms. A meeting for all participants and collaborators will be held to present the findings. If indicated, the results will be used in a funding application for a future trial.

## TRIAL STATUS

Recruitment for this trial began in August 2019 and closed in December 2019. Baseline data were collected in December 2019 and January 2020, then the schools were randomised and informed of their group allocation. Intervention packs were delivered to schools in March 2020. The study was paused in March 2020 because of the COVID-19 pandemic. At the time of submission (14 September 2020), we have obtained FREC and HRA permission to restart the study, with minor amendments to the protocol in line with COVID-19 precautions. Follow-up data will be collected in January 2021. The COVID-19 amendments will be reported with the results.

**Acknowledgements** The authors are very grateful to Mai Baquedano for her help and innovation with the REDCap database, to Cassandra Wye for the stories and learning resources and to Professor Innes Cuthill for advice and support with measuring visual clutter.

**Contributors** CW: designed and led the study, with input from all authors; AP and TG: led and designed the process evaluation; PS: provided governance support; RW: managed the databases; KB: designed and led the economic evaluation; DG: provided statistical support. All authors contributed to and approved the final manuscript.

**Funding** This study presents independent research funded by the National Institute for Health Research (NIHR) Senior Fellowship award SRF_2015_08_005.

**Disclaimer** The views expressed are those of the author(s) and not necessarily those of the NHS, the NIHR or the Department of Health and Social Care.

**Competing interests** None declared.

**Patient consent for publication** Not required.

**Provenance and peer review** Not commissioned; externally peer reviewed.

**ORCID iDs**
Cathy Williams http://orcid.org/0000-0002-9133-2021
Anna Pease http://orcid.org/0000-0002-3472-1047

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
