## [Reviewer comments · BMJ Open]

ARTICLE DETAILS

TITLE (PROVISIONAL)	Improving outcomes for primary school children at risk of cerebral visual impairments (The CVI Project): Protocol of a feasibility study for a cluster-randomised controlled trial and health economic evaluation
AUTHORS	Williams, Cathy; Pease, Anna; Goodenough, Trudy; Breheny, Katie; Gaunt, Daisy; Sinai, Parisa; Watanabe, Rose

VERSION 1 – REVIEW

REVIEWER	Sonja Alimovic University of Zagreb, Faculty of Education and Rehabilitation Sciences
REVIEW RETURNED	03-Oct-2020

GENERAL COMMENTS	This study has a very interesting topic. There are many studies now investigating CVI, symptoms, assessment, potential adaptations in order to gain better visual functioning in children, but this would be interesting to see the influence of intervention on the QoL in children, parents, and also satisfaction of teachers when they will know more. However, I am a bit afraid that the material which is a bit “overall” is not going to be very helpful to all teachers and all children. As I understood, they are going to get introduction in CVI, but every child with CVI has different visual functioning. Wouldn't it be better to give teachers concrete information regarding children? Perhaps they already got them from their special teachers working with students with visual impairment (VI teacher). I do not see that any of VI teachers are going to be involved in this study, and I think they should be. While reading the project protocol, at first, I thought the objective is to assess the improvement of families and children's quality of life (since the outcome measures are “QoL for children, parents...”) Nevertheless, from the part describing analysis, it is not going to be so. The researchers are going to analyse the recruitment, the percentage of collected data etc. But they are not going to analyse the difference between QoL before and after the intervention. So, there is a question, why to use so many questionnaires, only to assess the number of families which are going to get involved in the study. Another issue is, why to involve only 5% in ophthalmology assessment? I understand it is expensive to do the ophthalmology assessment to all children, but this way some children who really need the assessment could be left out. It does not seem very ethical. So, how the researchers are going to decide who will enrol in this assessment and who is not going to if they are going to have similar problems. Furthermore, children with CVI have problems in visual functioning (not due to ocular problems), and the functional visual assessment is going to be done only once, by the end of intervention
--

	period. Which data do researchers want to collect through this functional visual assessment and why. Wouldn't it be better to do it at the beginning, so they could give specific information on child's visual problems to teachers during intervention period? It seems to me that researchers have a lot of objectives in one study, and they have very good main goal, but written like this it is a bit mixed up. I would recommend to chose outcome measures based on the objective. If they want to analyse the recruitment, perhaps they do not need to have so many questionnaires. Perhaps parents could fill up the questionnaires, but they will make no change in everyday activities concerning the child's vision problems. Or, if they want to do all those questionnaires before and after, then they should analyse the difference between them to analyse potential change. So, I would recommend to make this study protocol a bit more simple, focus on one thing at a time, or to describe in more details: why to use all those questionnaires before and after, why to do ophthalmology and functional vision assessment and especially, why to do them by the end of intervention, and not at the beginning. I would like to know what is going to happen if there will be more than 5% of children in need for ophthalmology assessment? I would also like researchers to describe what are going to do in if the schools are going to share the materials (quote: intervention group schools will be asked not to share the intervention materials however it is recognised that contamination between arms may occur and the PE will investigate this.), after they investigate it. Are they going to exclude the school from the study, what if all of them will share the materials?
--	--

REVIEWER	Niranjan Pehere L V Prasad Eye Institute, Liberia
REVIEW RETURNED	05-Oct-2020

GENERAL COMMENTS	Dear authors: This is a brilliant attempt to address a very important question in CVI- does intervention really help... congratulations! I would like to draw your attention to a few things in the paper:  1.It's too long a paper, please tighten it, making it precise. 2. There are many sentences that are too long- you may consider making them shorter to make them reader friendly. 3. The intervention is basically about empowering teachers with some basic knowledge about CVI, which is not given to the control schools. The issue with this is- Logically, the knowledge about CVI is definitely going to help teachers detect students with those issues. Then does this really need an RCT? Are we not denying an obvious advantage to the control group?  4. Setting a 5% cut off to the number of children that will be seen in the clinic is also an ethical issue. If we are going to detect cases by a study intervention, we need to find ways to take care of them, which may be more than 5%. It is possible that quite a few non-CVI kids may be detected through this intervention and total number of children found to have learning
---

	related issues may be more than 5%. 5. If I may suggest, following study design may be ethically more appropriate- Post teachers' training, few children will be detected to have issues and referred to the CVI clinic. Some children will follow-up, some will not for various reasons. Those who do not, become a control group without any ethical issues. You may like to compare QoL among those two groups, post intervention. And intervention here will be the strategies suggested by the CVI clinic. The need for evidence is for the effectiveness of intervention strategies, and denying strategies to a group by RCT does not seem a ethical thing to do.
--	--

REVIEWER	Kathleen Vancleef University of Oxford, United Kingdom I have received support from the authors in preparing and running a patient and public involvement session for my own research.
REVIEW RETURNED	07-Oct-2020

GENERAL COMMENTS	Dear editor, Dear authors, Thank you for giving me the opportunity to review your manuscript. You describe the protocol for a feasibility randomised control trial with the long term aim to evaluate the effectiveness of an intervention that can potentially improve support of children with Cerebral Visual Impairment in an educational setting. Data collection for the study is ongoing. I will therefore limit my recommendations to clarifications and inclusion of methodological details. Background  - I very much agree that there is no consensus yet on how CVI should be defined.  o Can you therefore please clarify how CVI was defined and how the diagnosis was reached for the prevalence that you report on Line 44 of page 6? o It is not clear to me how you are going to evaluate effectiveness in the follow-up RCT when the protocol does not allow you to confidently differentiate between children with and without CVI. Surely, given that the intervention is specifically designed for children with CVI, you expect a higher benefit for this group, right? So, you would want to differentiate both groups. Can a formal diagnosis be made based on the orthoptic assessment after referral? It would be helpful if you could clarify this in the manuscript. - Line 27-31: It might be worth pointing out that the developmental delay you observed in children with CVI symptoms is not necessarily due to their CVI, but might be caused by other factors such as the disabilities of mental health issues you report in the previous sentence, unless you have controlled for this in your analysis? - When introducing the intervention on page 7, it would be helpful to the reader to start with describing the aims of the intervention. I was left wondering if it was to remit underdiagnosis, improve referral or improve educational support. - The description of the intervention on page 7 (line 35-42) is rather vague. The school component is described in more detail on p12 and it might be worth referring the reader to that section. Do teachers get any guidance or a set of criteria on which children
---

should be referred to an eye clinic? I am certainly missing a more detailed description of the eye clinic component: which 'simple vision tests' were provided and what was included in the 'advice sheets' See also my question above on whether a diagnosis would be made or not. Would it be possible to add some representative examples of the materials as supplementary material (possibly only publically released after data collection has been completed)?

- Thank you for sharing the outcomes of your patient and public involvement sessions. That is a very valuable component of your research. I am interested to hear if you have consulted any teachers or orthoptists and what their recommendation for the protocol were.

- It seems that the objectives of the study do not completely match up with the reported outcomes measures and analyses. For instance, it is not clear to me how you will investigate the reasons for non-participation, predictors of non-participation, how you will determine which recruitment methods yield the best result, evaluate quality of the data and the retention in referrals. Can you please provide additional information in the analysis section or adjust the objectives?

Methods and analysis

- P9, line 3: It seems there is a discrepancy between your description of eligible schools here (all mainstream primary schools), and Figure 1 (only state-funded primary schools). Can you please clarify, also for a reader unfamiliar with the UK school system?

- P9, line 6: this sentence seems incomplete

- P9, line 10: can you please explain the abbreviation SENCo?

- P9, line 50-53: Can you please explain why collection of this personal information is essential? Is it being used in the economic evaluations?

- P 10, line 5: It would be helpful for the reader to hear about a few examples of COS that you have identified. Can you please summarise your previous work in a few sentences?

- Table 1: Is it possible to attach the family expense questionnaire that was designed for the study?

- I feel it would be helpful if a similar table was provided with the data collected in the eye clinic. It is currently unclear to me what information is collected at that stage and how it will be used in the analyses.

- P12, lines 5-8: Did you coordinate agreement of schools and POs in one area or could participating schools refer children to a PO outside there area if their local PO did not take part in the study?

- P12: Was there an incentive for the school to engage with the resources (powerpoint, online links, strategies, referrals)? It seems to put quite a demand on their time and potentially require them to make changes to their classroom and teaching practice. How do you keep them motivated?

- P12, line 47: Can you please explain 'intervention Logic Model'?

- P12, line 59-60 and p13 line 9-15: Can you please provide more details on how these data will be collected. Is it included in any of the questionnaires you list in Table 1?

- P13, line 52: Your estimate of 95% of questionnaires to be returned sounds rather ambitious to me, is this based on your previous experience with running similar studies on primary schools? If so, it might be worth adding this information to reassure the reader that 95% is a realistic estimate.

- P13, line 60: It is of course good practice to have an analysis plan in place before data collection, so the more details you can give here the more confident a reader will be about suitability of your protocol to address your research questions. You might want to rephrased the sentence on line 60.

	- Figure 1: It is unclear to me how data will be collected from children: will this be done with the teacher or will a researcher visit the school? - Figure 2: I missed a citation of the figure in the text. I was also wondering if any data will be collected on how referrals proceed through this pathway. For instance, how many GPs decide not to send the letter on and what are potential reasons for this? Your sincerely Dr Kathleen Vancleef
--	--

VERSION 1 – AUTHOR RESPONSE

Reviewer 1 (SA)

1. Wouldn't it be better to give teachers concrete information regarding children?

For children who are referred, the teachers will get concrete information from the clinic. However for the rest of the children, it is not feasible in the real world that all would have a detailed assessment and we are aiming to be pragmatic and only suggest interventions that might be possible in day-to-day life.

2. I do not see that any of VI teachers are going to be involved in this study, and I think they should be.

We have now explained that our local QTVIs have been closely involved and reviewed and contributed to the intervention pack materials, especially the universal and targeted strategies. We mention this in the first para, P5.

3. The researchers are going to analyse the recruitment, the percentage of collected data etc. But they are not going to analyse the difference between QoL before and after the intervention. So, there is a question, why to use so many questionnaires, only to assess the number of families which are going to get involved in the study.

We have now explained more clearly that this is a feasibility study, rather than a definitive trial (Abstract background; last para P5, Objectives section P6 and Analysis section P10-11). We need to report on the recruitment and data collection as they are important outcomes for a feasibility study (as described in the MRC guidance). We will however analyse the QoL data and look to see if there is any sign of an effect of the intervention - and we will use these data to help us determine how large the definitive trial will need to be in order to show a statistically significant effect, if present. We have included questionnaires to capture the views of the children, the teachers and the parents as all are important and we will use the data from this feasibility study to check that this number of questionnaires is acceptable to participants.

4. Another issue is, why to involve only 5% in ophthalmology assessment?

A limit on the number of was requested by the Ethics committee, in order to avoid overloading the GPs and/or the eye clinic. We have explained this in the Ethics section (2nd section P12)

5. Which data do researchers want to collect through this functional visual assessment and why. Wouldn't it be better to do it at the beginning, so they could give specific information on child's visual problems to teachers during intervention period?

We have suggested to the eye clinics that they use the protocol we used in our Prevalence study, to identify impairments in acuity, field, motility or visuocognitive skills. We agree ideally the children

would attend the eye clinic as early as possible during the intervention period. However, part of what we need to find out is whether the referral process we have suggested is practicable and how long it takes and we will ask teachers and parents and eye clinic staff explicitly about the referral process, during the PE. We will use this information to optimise the referral process.

6. I would like to know what is going to happen if there will be more than 5% of children in need for ophthalmology assessment?

We will ask the schools about this and will report if they wanted to refer more than 5% and use this information to increase the number of referrals planned for in the future trial. The children's GPs can refer as many children as they want, and teachers and parents will be encouraged to ask for referral, if needed. We describe this on P5 first para (referrals as part of the intervention) and P6 first para (outcomes)

7. I would also like researchers to describe what are going to do in if the schools are going to share the materials (quote: intervention group schools will be asked not to share the intervention materials however it is recognised that contamination between arms may occur and the PE will investigate this.), after they investigate it. Are they going to exclude the school from the study, what if all of them will share the materials?

No, we will not exclude or censure in any way schools or parents if there is sharing of the study materials. If there is widespread sharing this will be an important outcome from this feasibility study and will indicate that randomising the schools into control and intervention groups the way we have suggested, is not practicable and we will change the design of the future trial.

Reviewer 2 (NP)

1. Too long and please shorten sentences

We have simplified the language and shortened sentences throughout. The revised paper is shorter by 353 words and is well within the journal limits for a Protocol paper. We are happy to be guided by the Editor if further shortening is required.

2. Then does this really need an RCT? Are we not denying an obvious advantage to the control group?

Our aim is to provide evidence regarding whether this particular intervention is effective at helping children with CVI. An RCT is an established way to test a new intervention but we agree it is possible may not be suitable in this case, if for instance there is widespread contamination between the arms once the existence of CVI is explained. This is one reason why this feasibility study is useful, to see if an RCT would be acceptable to participants.

3. Setting a 5% cut off to the number of children that will be seen in the clinic is also an ethical issue. As we have now explained, we were requested to do this by the Ethical committee we will emphasise to teachers and parents that their GP can still refer them for a vision assessment, outside the study.

4. The need for evidence is for the effectiveness of intervention strategies, and denying strategies to a group by RCT does not seem an ethical thing to do.

The families in our advisory group recommended that the study materials and information about CVI be made available to the control group at the end of the study and we will do this, as we now explain (PPI section P12). The families and the Ethical committee raised no ethical concerns about the study.

Reviewer 3 (KV)

1. Can you therefore please clarify how CVI was defined and how the diagnosis was reached for the prevalence that you report on Line 44 of page 6?

The prevalence we quote is for “CVI-related vision problems” rather than “CVI” per se, because we agree with the reviewer that there is no consensus regarding the type of severity of problems necessary to warrant a formal diagnosis of CVI. We explain this in the introduction (paras 1 and 3, P4).

2. Surely, given that the intervention is specifically designed for children with CVI, you expect a higher benefit for this group, right? So, you would want to differentiate both groups. Can a formal diagnosis be made based on the orthoptic assessment after referral?

We agree it would be ideal to identify all the children with CVI in each group and compare the outcomes after exposure to the intervention or not. However, there are no questionnaire tools which will do this accurately in a school setting. We did include the teacher-report Gordon Dutton 5Qs, but our earlier study suggested this is not very sensitive and would identify only 13% of children with CVI-related vision problems.

If we examined all children at the study start to identify those who fitted a definition of CVI, or all those with CVI-related vision problems, it would then be unethical not to offer help to identified children and so we would not have a control group. Certainly we would expect children examined in the eye clinic could have a diagnosis, but this will not apply to the control group

We will therefore examine the usefulness of regarding the SEND children as an already-identified (by the schools) sub-group that is enriched with children who have CVI or at least CVI-related vision problems. In our prevalence study we found that 41% of SEND children had one or more CVI-related vision problems. We have explained this more clearly in the analysis section on P10-P11.

3. developmental delay you observed in children with CVI symptoms is not necessarily due to their CVI, but might be caused by other factors such as the disabilities of mental health issues you report in the previous sentence, unless you have controlled for this in your analysis?

We agree that just because a child has a need for extras educational help (as determined by the school) and has vision problems, that the two are not necessarily causally related. This is indicated by the results of our prevalence study which were that “extra help” was poorly specific for CVI-related vision problems. We have added this in the last para of introduction on P4.

4. to start with describing the aims of the intervention. I was left wondering if it was to remit underdiagnosis, improve referral or improve educational support.

The aim is to improve outcomes (QoL, learning) for children with CVI. The pathways by which this might be achieved include all factors mentioned by the reviewer, plus potentially others, and the aim of our Process Evaluation is to gain information on this. For example does teacher understanding of CVI change? Do they adapt their teaching methods in response to the intervention? These will be outcomes from the PE. We have explained this at the end of the section on the development of the intervention (P5-6).

5. Do teachers get any guidance or a set of criteria on which children should be referred to an eye clinic?

Yes they are given specific criteria in the CVI information pack.

6. I am certainly missing a more detailed description of the eye clinic component: which 'simple vision tests' were provided and what was included in the 'advice sheets'.

The tests are described in detail in our prevalence paper and we can now reference this as it is in press. The advice sheets are collation of existing resources eg TeachCVI, Gordon Dutton's strategies, with additional input from our QTVIs. We have not yet included them with this paper as the study is ongoing but at the end of the study, they will be available on request on our website. We now mention this and give the website address, (P5)

7. I am interested to hear if you have consulted any teachers or orthoptists and what their recommendation for the protocol were.

We have described now the input from our advisory groups, in the PPI section. The professionals group includes a teacher, an orthoptist and a SENDCo. We explain their input and that of the parents, in our PPI section on P12.

8. how you will investigate the reasons for non-participation, predictors of non-participation, how you will determine which recruitment methods yield the best result, evaluate quality of the data and the retention in referrals. Can you please provide additional information in the analysis section or adjust the objectives? Retention of schools once recruited will be expressed as the number of schools who withdraw from the study, after signing the MOU.

Reasons for non-participation will be noted from phone calls and emails with schools enquiring about the study but then deciding not to participate. We will regard the best recruitment methods (eh flyers or emails) as the one with the highest yield of schools who agree to take part. Quality of data returns will be evaluated as proportion of missing data. This has been clarified in the analysis section (P10-11) and in the objectives on P6.

9. It seems there is a discrepancy between your description of eligible schools here (all mainstream primary schools), and Figure 1 (only state-funded primary schools).

We have added "state-funded" to the description of eligible schools

10. P9, line 6: this sentence seems incomplete

- P9, line 10: can you please explain the abbreviation SENDCo?
- P9, line 50-53: Can you please explain why collection of this personal information is essential? Is it being used in the economic evaluations?

We have corrected the sentence in the original document P6, line 6 - in the section on Inclusion criteria (P7).

We have explained the abbreviation SENDCo in the section on Exclusion criteria (P7)

We will collect data on free school meals as a proxy for area deprivation so that we can describe the demographic characteristics of the participants. We will collect data on whether the child is having extra educational help as we expect this group to be enriched with children with CVI, as explained in the introduction (P4) and analysis sections (P10-11).

11. t a few examples of COS that you have identified. Can you please summarise your previous work in a few sentences?

As the data are not yet published we are not able to give more details of the Core Outcome set that was produced

12. Is it possible to attach the family expense questionnaire that was designed for the study?

We have now included this as an appendix.

13. similar table was provided with the data collected in the eye clinic. It is currently unclear to me what information is collected at that stage and how it will be used in the analyses.

We have now referenced the paper where we describe the tests we will suggest the eye clinic use. The results will be used to indicate whether the teachers refer children who do indeed have CVI-related vision problems, or not. We will use this to help understand whether this is an effective way of getting assessments for children with CVI. We now explain this in the Analysis section (P11)

14. Did you coordinate agreement of schools and POs in one area or could participating schools refer children to a PO outside there area if their local PO did not take part in the study?

The areas were chosen because the local POs agreed to take part, as the schools would not have been able to ask the GP to refer to a clinic outside of their area. We describe this on P7 in the section on Setting

15. Was there an incentive for the school to engage with the resources (powerpoint, online links, strategies, referrals)?

No, other than the explanations given in the Study information leaflet. The degree to which they actually engaged will be examined in the Process Evaluation.

16. P12, line 47: Can you please explain 'intervention Logic Model'?

A Logic model is a diagram which shows the factors and pathways hypothesised to underlie the intervention. This is described in detail in the PE.

17. P12, line 59-60 and p13 line 9-15: Can you please provide more details on how these data will be collected. Is it included in any of the questionnaires you list in Table 1?

Yes, the "quantitative data" refers to the responses to the questionnaires listed in Table 1. Recruitment data will be stored on an Excel sheet by the study administrator.

18. Your estimate of 95% of questionnaires to be returned sounds rather ambitious to me, is this based on your previous experience with running similar studies on primary schools? If so, it might be worth adding this information to reassure the reader that 95% is a realistic estimate.

We now refer to our earlier study in which this was the response rate we achieved.

19. It is of course good practice to have an analysis plan in place before data collection, so the more details you can give here the more confident a reader will be about suitability of your protocol to address your research questions. You might want to rephrase the sentence on line 60.

As this is a feasibility study, rather than the definitive trial, we have removed the sentence as we agree it may cause confusion. The future trial would have a formal analysis plan as part of the future funding application.

20. Figure 1. It is unclear to me how data will be collected from children: will this be done with the teacher or will a researcher visit the school?

The children will complete self-report questionnaires in their classroom, supervised by school staff. We have clarified this in Table 1.

21. Figure 2: I missed a citation of the figure in the text. I was also wondering if any data will be collected on how referrals proceed through this pathway. For instance, how many GPs decide not to send the letter on and what are potential reasons for this?

We have added a citation to Figure 2 on P7. We will ask the schools to record how many children they refer and the eye clinic to record how many appointments they offer. We have no plans to interview GPs and will be unable to describe their reasons for not referring a child. We have added this to the study limitations.

VERSION 2 – REVIEW

REVIEWER	Sonja Alimović University of Zagreb, visual impairment
REVIEW RETURNED	28-Feb-2021
GENERAL COMMENTS	Dear authors, I am very glad that you have accepted the suggestions, and now you have more clear protocol on very interested topic. Not only that topic is scientifically interesting, but it is also very important for practice, for students, parents, and teachers. Hope to see your results published soon. Kind regards
REVIEWER	Niranjan Pehere LV Prasad Eye Institute
REVIEW RETURNED	31-Jan-2021
GENERAL COMMENTS	Dear authors: Thanks for getting back with all the queries answered well. This is a much needed study
REVIEWER	Kathleen Vancleef University of Oxford, United Kingdom
	I have received support from the authors in preparing and running a patient and public involvement session for my own research.
REVIEW RETURNED	10-Feb-2021
GENERAL COMMENTS	I am happy with how the authors have addressed my comments and answered my questions.